

# Soil moisture droughts in Belgium during 2011–2020 were the worst in five decades

Katoria Lekarkar[1][*], Oldrich Rakovec[2], Rohini Kumar[3], Stefaan Dondeyne[1,4] and Ann van Griensven[1,5]

[1]Department of Water and Climate, Vrije Universiteit Brussel, Pleinlaan 2, 1050, Brussels, Belgium.

[2]Faculty of Environmental Sciences, Czech University of Life Sciences Prague, Praha-Suchdol, Czech Republic.

[3]UFZ-Helmholtz Centre for Environmental Research, Permoserstraße 15, 04318, Leipzig, Germany.

[4]Gembloux Agro-Bio Tech, University of Liège, Pass. des Déportés 2, 5030, Gembloux, Belgium.

[5]Water Science & Engineering Department, IHE Delft Institute for Water Education, 2611 AX, Delft, The Netherlands.

[*]Corresponding author: katoria.lesaalon.lekarkar@vub.be;

## Abstract

In recent years, Belgium has experienced a sequence of intense droughts with wide-ranging impacts across multiple sectors. Determining whether these events are unprecedented or within natural variability requires indicators that properly diagnose drought. Root-zone soil moisture is a suitable indicator because it integrates meteorological forcings with land-surface processes. In Belgium, however, operational monitoring relies mainly on precipitation-based indices and lacks long-term in-situ soil-moisture observations, leaving uncertainty about whether these indices capture the persistence of root-zone drought. To address this gap, we reconstructed daily root-zone soil-moisture dynamics over Belgium for 1970–2020 using the mesoscale Hydrologic Model (mHM), placing recent droughts in historical context and evaluating the adequacy of precipitation-based indicators for representing drought conditions. Our analysis shows that droughts in 2011–2020 were unprecedented in both duration and severity over the past five decades.Between 2011 and 2020, the country experienced a cumulative three years of drought (non-consecutive),





representing 30% of the decade, more than double the cumulative duration
in each decade from 1981–2010 and about 1.5 times that of 1971–1980.
We further find that the Standardized Precipitation–Evapotranspiration Index
(SPEI), currently used operationally as a proxy for agricultUral droughts in
Belgium, underestimates the persistence of root zone droughts because it
does not explicitly account for land-surface memory. Thus, by including soil
moisture monitoring in drought assessment, residual stresses on agriculture
and subsurface water which can persist long after meteorological condi-
tions have normalized can still be detected. This gives decision-makers a
more realistic understanding of droughts and how to respond proportionately.

**Keywords:** Mesoscale, climate variability, drought persistence, agricultural drought
monitoring

# 1 Introduction

Belgium has faced a succession of hugely consequential droughts in recent years. These
droughts led to declined crop yields, increased water scarcity and restricted water abstractions,
disrupted navigation on inland waters and caused economic losses running into millions of
Euros (Tröltzsch et al., 2016; De Ridder et al., 2020). Between January and April 2011, Bel-
gium had only received less than 50% of the climatologically expected rainfall by that time
of the year (European Commission, Joint Research Centre, 2011). In 2018-2019, a multi-year
drought characterized by rainfall deficits and record-breaking temperatures swept through the
country, causing significant economic costs across different sectors (Bastos et al., 2020). In
the Flemish region (the northern part of the country), the event reduced potato production by
31% leading to a 23% surge in prices. Sugar beet production fell by 13% and cereal yields
reduced by 10%. These led to farmers submitting claims of about €150 million to the Flemish
Disaster Fund to compensate for losses from the drought (De Ridder et al., 2020). According
to the agency in charge of inland water in Flanders (De Vlaamse Waterweg nv), inland navi-
gation suffered economic losses of more than €300 million due to low water levels during the
2018-19 drought. In July 2019, a temperature record of 39.7°C was measured, which marked
the most intense heatwave ever recorded in the country at the time (Chini, 2022). Soon after,
in 2022, another drought hit Belgium, affecting 53.4% of the country, more than ten times the
long-term average impacted area of 4.6% between 2000 and 2020. According to the Coper-
nicus Climate Change Service (https://climate.copernicus.eu/esotc/2022/drought), surface soil
moisture in Europe throughout 2022 was the second lowest in the last 50 years, sustained by
higher-than-average temperatures and a sequence of heatwaves that started in spring and con-
tinued throughout summer. Of all European countries, Belgium was the second most affected
country in terms of the proportion of area impacted by the drought (European Environment



Agency, 2023). By March that year, water levels in half of the groundwater wells in Flanders were very low for that time of the year. By May, this number had increased to two-thirds (Walker, 2022). In July, rainfall in the country was the lowest in 137 years (since 1885), with an average rainfall of 5 mm across the country. The drought caused significant crop damage, and the Flemish government subsequently declared the drought a disaster, which paved the way for farmers to seek compensation for crop losses. Evidently, these recent drought events are well documented. In order to contextualize their magnitudes and severity, it is essential to reconstruct historical drought occurrences over a sufficiently long climatological period. Such a long-term perspective is necessary to determine whether recent droughts are unprecedented extremes or if they fall within the range of natural climate variability.

Belgium has an extensive network of precipitation, river discharge and groundwater monitoring stations which provided the basis for monitoring hydrological and meteorological droughts. This data underlies the drought indices found in dedicated platforms for tracking and communicating the evolution of droughts across the country (e.g. https://www.meteo.be/en/weather/forecasts/drought, https://vmm.vlaanderen.be/water/droogte). However, due to the lack of long-term observations of soil moisture in the country, the extent of agricultural droughts is presently evaluated with the Standardized Precipitation Evaporation Index (SPEI)(Vicente-Serrano et al., 2010) which expresses anomalies in the climatic water balance, that is, precipitation minus potential evapotranspiration. The nationwide drought conditions are reported through https://www.meteo.be/en/weather/forecasts/drought. Although useful, precipitation- and temperature-based drought indices are constrained for their limited ability to fully represent agricultural drought conditions. Firstly, these indices do not explicitly account for the vertical distribution of water within the root zone that supports plant growth, nor do they reflect the complex interactions between soil moisture and vegetation across different stages of plant development and are thus inadequate to represent extreme water shortage that would lead to biomass and crop yield reduction (Sheffield et al., 2004; Mishra and Singh, 2010; Samaniego et al., 2013). While soil moisture may exhibit direct link to precipitation at monthly timescales, soil moisture responses can be nonlinear at shorter timescales, particularly during dry conditions. Soil moisture also has a memory effect that can lag precipitation anomalies by days to months and in turn prolong the persistence and severity of drought (Bonan and Stillwell-Soller, 1998; Nicholson, 2000; Wu et al., 2002; Seneviratne et al., 2006). Accordingly, developing indices based on soil moisture offers a more reliable indicator of agricultural drought as soil moisture integrates the effects of antecedent precipitation, plant water uptake through transpiration, and the increasing persistence of soil wetness with soil depth (Wu et al., 2002; Sheffield et al., 2004).



The goal of this study is therefore to perform a retrospective high-resolution reconstruction of root zone soil moisture to perform a first-of-its-kind assessment of soil moisture droughts in Belgium over the five decades between 1970 and 2020. We aim to characterize major droughts that have occurred over this period by clustering soil moisture anomalies using thresholds that capture the spatiotemporal characteristics of identified events and rank them based on their magnitude, spatial extent and duration, and evaluate how drought patterns in the country have evolved over the five decades. In our study we use the mesoscale hydrological model (mHM) driven by offline meteorological forcings to simulate soil moisture conditions and derive grid cell-level statistical distributions for characterizing the spatial and temporal patterns of agricultural drought over Belgium. To evaluate the correspondence between SPEI and soil moisture-based anomalies to represent agricultural droughts, we compare SPEI at different accumulation periods to a soil moisture index (SMI) (Samaniego et al., 2018), derived from monthly percentile ranking of soil moisture fields, during selected major drought events.

## 2 Methodology

## 2.1 Study domain

Belgium is located in Western Europe covering an area of 30,528 km$^2$, varying in topography from sea level along the North Sea coast to 700 m in the Ardennes-Eifel massif in the south eastern parts (Figure 1) (Meersmans et al., 2016; Sousa-Silva et al., 2016). The country experiences a warm temperate maritime climate (Köppen-Geiger Cfb) strongly modulated by the warming effect of the North Atlantic Drift (Erpicum et al., 2018; Beck et al., 2023). Data from the Royal Meteorological Institute of Belgium (RMI) shows that mean annual temperature ranges between 13 and 17 $^0$C, varying spatially with elevation and distance inland. Winters are generally mild, with December–January lows dipping under 5$^0$C but rarely below freezing conditions for prolonged periods. Winters are colder in the Ardennes region due to a weaker maritime influence and higher elevation. Summers are moderately warm with July highs peaking around 18$^0$C although extremes above 30$^0$C have occurred in recent years. The country receives an annual average precipitation of about 800 mm which varies between 700 mm in the western low lying regions, up to 1400 mm in the Ardennes where precipitation is enhanced by orographic effects (Erpicum et al., 2018). Temporally, rainfall is fairly evenly distributed throughout the year (Figure 1), with seasonal patterns dominated by summer convective storms and winter frontal systems (Brisson et al., 2011; Goudenhoofdt and Delobbe, 2013; Journée et al., 2015).

Land cover in the country is predominantly agricultural (44%), dominated by croplands and animal husbandry. Cultivated areas dominate the central loamy belt and the northwest of the country while the coastal polders typified by heavy soils, are more suited for animal-based



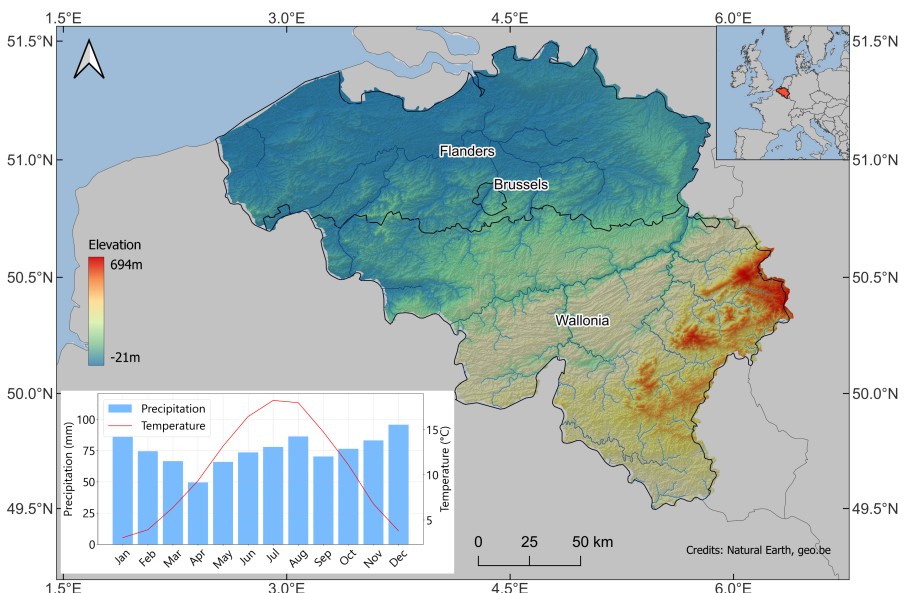

**Fig. 1**: Topographic map of Belgium. The Ardennes region is distinguishable by its high elevation in the south east. Monthly mean precipitation and temperature in the inset plot are derived from data provided by The Royal Meteorological Institute of Belgium for the climatological period 1994-2023.

farming (Beckers et al., 2018, 2020; Statbel, 2025a). Forests cover about 23% of the territory (just over 700,000 hectares) with 79.8% in the Walloon region, 19.9% in Flanders and 0.3% in the Brussels-Capital (Sousa-Silva et al., 2016; Royal Forestry Society of Belgium, 2025). Most of the lowland forests are dominated by broad-leaved tree species with clusters of coniferous forest plantations in the north east. In the Ardennes, forests form a mixed broadleaved–coniferous complex in the foothills, gradually transitioning to conifer-dominated stands at higher elevations(Royal Forestry Society of Belgium, 2025; Statbel, 2025a). Built-up and urbanized areas account for about 20% of the land with most cities dating back to the Middle Ages. The average population density of the country is 385 inhabitants/km$^2$ (Beckers et al., 2020; Statbel, 2025b).

## 2.2 The mesoscale Hydrologic Model

We used the mesoscale Hydrologic Model (mHM; Samaniego et al., 2010; Kumar et al., 2013) (version v-5.13.2-dev0) to simulate domain-wide root zone (0-2 m) soil moisture conditions and streamflow, which we used as an additional hydrologic constraint for validating basin-scale



hydrology at major outlets. mHM is a spatially distributed hydrological model based on numerical representations of dominant hydrological processes. The model is driven by hourly to daily meteorological forcings, which include precipitation, temperature, and potential evapotranspiration, and accounts for major hydrological processes like snow melt and accumulation, canopy storage, evapotranspiration, surface runoff and flood routing, three-layer soil moisture content, and subsurface storage. To represent spatial variability of inputs and state variables, the model uses three different spatial resolutions, namely (in order of fine to coarse resolution); Level-0 ($L_0$: small scale morphology) to represent the main terrain features, geological features, land cover and soil properties; Level-1 ($L_1$: mesoscale hydrology) to represent the dominant hydrological processes; and Level-2 ($L_2$: large scale meteorology) to describe the variability of meteorological forcings. mHM links model parameters at $L_1$ to their corresponding ones at $L_0$ using multiscale parameter regionalization (MPR; Samaniego et al., 2010). This technique uses non-linear transfer functions that couple catchment characteristics with global (calibration) parameters to regionalize model hydrologic parameters at $L_0$ and link them to their corresponding values at $L_1$ using upscaling operators such as arithmetic mean, geometric mean, and harmonic mean (MPR; Livneh et al., 2015). With this technique, mHM overcomes the problem of overparameterization and model equifinality (Samaniego et al., 2010, 2011; Kumar et al., 2013; Samaniego et al., 2013). mHM has been successfully used in multiple studies at scales ranging from river basins (Zink et al., 2017; Dembélé et al., 2020; Demirel et al., 2024; Banjara et al., 2025), country level (Samaniego et al., 2013; Rakovec et al., 2019; Boeing et al., 2022) up to continental-scale (Samaniego et al., 2018; Moravec et al., 2019; Kumar et al., 2025) and global studies (Řehoř et al., 2025; Shrestha et al., 2025).

### 2.2.1 Input data

Without long-term in situ soil moisture within Belgium to validate the soil moisture output of mHM, we expanded the model domain to cover parts of France, Germany and The Netherlands where soil moisture observations are available from the International Soil Moisture Network (ISMN) (Dorigo et al., 2021), shown in Figure 2. We subsequently forced the model with daily fields of precipitation and temperature from the ENSEMBLES gridded dataset (E-OBS) version 30.0e (Cornes et al., 2018), which covers the entire modelling domain. E-OBS is a daily land-only gridded observational dataset over Europe which blends station network time series from the European National Meteorological and Hydrological Services or other sources and is provided with spatial resolutions of $0.1^0$ and $0.25^0$. Our setup uses the $0.1^0$ resolution product (access url: https://cds.climate.copernicus.eu/datasets/insitu-gridded-observations-europe?tab=download, last accessed March 2025). Since



E-OBS does not provide potential evapotranspiration data, we generated this from the E-OBS minimum and maximum temperature using the method of Hargreaves and Samani (1985).

The morphological datasets for the model originate from different sources namely; LAI maps from Global Inventory Modeling and Mapping Studies (GIMMS) (Cao et al., 2023), DEM from the Shuttle Radar Topography Mission (Farr et al., 2007), land use data from Corine Landcover (https://land.copernicus.eu/en/products/corine-land-cover), soil texture and bulk density data from the Harmonized World Soil Database (Nachtergaele et al., 2023), and geology datasets from the Global Lithological Map Database (Hartmann and Moosdorf, 2012), accessed from the url: https://www.geo.uni-hamburg.de/geologie/forschung/aquatische-geochemie/glim.html (last accessed February 2025). To ensure the spatial consistency required by mHM, we prepared all $L_0$ datasets at 0.001953125° (1/512°), bilinearly coarsened the $L_2$ meteorological data to 0.125° (1/8°), and set the resolution of $L_1$ to 0.03125° (1/32 °), these are summarized in Table 1. We then run the model from 1965 to 2023, including a warm-up period of 5 years at the beginning.

**Table 1**: Summary of data sources

| Dataset | Resolution (degrees) | Input format | Source |
| --- | --- | --- | --- |
| Meteorological data | 1/8 | NetCDF | RMI Belgium |
| Leaf Area Index | 1/512 | NetCDF | GIMMS |
| DEM | 1/512 | ASCII Grid | SRTM |
| Geology | 1/512 | ASCII Grid | Global Lithological Map Database |
| Land Cover | 1/512 | ASCII Grid | Corine Landcover |
| Soil texture | 1/512 | ASCII Grid | Harmonized World Soil Database |

### 2.2.2 mHM Soil Moisture simulation

mHM calculates water infiltration between soil layers using an exponential function that accounts for the nonlinearity of soil water retention (Samaniego et al., 2010; Livneh et al., 2015). Briefly, for a given soil layer, $k$, on pervious areas, the infiltration $I_k$ into the layer is determined by the equation:

$$I_k = I_{k-1} * \left( \frac{\theta_k}{\theta_{sat,k}} \right)^{\beta_k} \qquad (1)$$

$I_{k-1}$ represents the infiltration from the previous layer $k-1$, $\theta_k$ is the soil moisture of layer $k$, $\theta_{sat,k}$ is the saturation moisture content for the layer, and $\beta_k$ is an exponential parameter that adjusts for the non-linear nature of soil moisture retention. Once infiltration is calculated, the




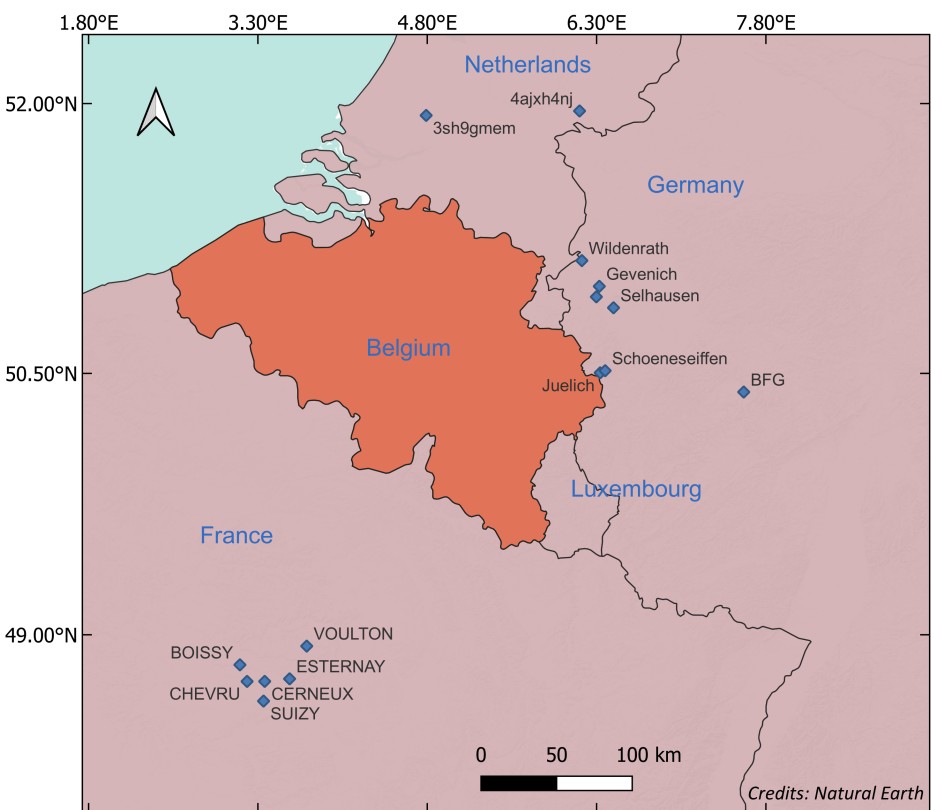

**Fig. 2**: Locations of ISMN stations (blue diamonds) used to validate mHM soil moisture

model updates soil moisture $\theta_t$ by adding the difference between the layer infiltration $I_t$ and actual evapotranspiration ($ET_t$) for the time step as;

$$\theta_t = \theta_{t-1} + I_t - ET_t \tag{2}$$

Actual evapotranspiration is calculated by reducing the potential evapotranspiration (PET) based on a soil moisture stress factor, $f_{SM}$, which varies depending on the soil moisture content.

$$ET = f_{roots} \cdot f_{SM} \cdot PET \tag{3}$$

$f_{roots}$ is the fraction of roots in the soil horizon and $f_{SM}$ is calculated using either the Feddes equation (Feddes, 1982):

$$f_{SM} = \frac{\theta - \theta_{pwp}}{\theta_{fc} - \theta_{pwp}} \tag{4}$$



or the Jarvis equation (after Jarvis (1989)):

$$f_{SM} = \frac{1}{\theta_{\text{stress-index-C1}}} \cdot \frac{\theta - \theta_{pwp}}{\theta_{sat} - \theta_{pwp}}$$  (5)

The model uses the MPR routine to compute the saturation moisture content, field capacity

($\theta_{fc}$) and wilting point ($\theta_{pwp}$).

### 2.2.3 Model evaluation

The accuracy and spatial representativeness of absolute soil moisture values are strongly

source-dependent (in situ or modelled), so direct comparisons between different datasets can

be misleading (Koster et al., 2009; Ford and Quiring, 2019). On one hand, simulated soil

moisture is highly dependent on the quality of meteorological forcings and the physical param-

eterisation of the model (Koster et al., 2009; Wang et al., 2011a; Nicolai-Shaw et al., 2015). On

the other hand, in situ measurements are highly localized to the sensor location and are affected

by the technology used by the sensor and the sufficiency of the calibration techniques (Peng

et al., 2025). From a drought analysis perspective, the real information value of soil moisture

is not in its absolute values but rather in its temporal variability metrics, such as anomalies

and seasonal variability of soil wetness (Koster et al., 2009). This information value is gener-

ally more consistent and transferable between different sources when soil moisture is suitably

normalised to have the same range and variability (Dirmeyer et al., 2004; Wang et al., 2011b).

Koster et al. (2009) show that if soil moisture from different sources differs only in their mean

and standard deviation, then standardizing each time series (as in Equation 6) would generate

nearly identical datasets of standard normal deviations ($\theta'$).

$$\theta' = \frac{\theta - \theta_m}{\sigma_m}$$  (6)

Where $\theta$ is the soil moisture at a given point and time of year, $\theta_m$ and $\sigma_m$ are the mean and

standard deviation of soil moisture, respectively, for the same point and time of year.

In our evaluation of the mHM soil moisture, we used this approach to analyze the level of

temporal agreement between the standard normal deviations of mHM and in situ soil moisture

from the corresponding depths at the selected ISMN stations (Figure 2).

For each in situ–modelled pair, we quantified the agreement in drought anomaly dynamics

by calculating the Pearson correlation coefficient ($r$). To obtain an overall agreement across all

sites, we first transformed the $r$ values to the Fisher $z$-scale ($z$ = arctanh(r)) to stabilize variance

and avoid bias from the nonlinear r-scale. The z-values were then averaged to obtain $\bar{z}$, and

finally back-transformed to yield $\bar{r}$ = tanh $\bar{z}$.

Prior to the comparison, we performed a quality check on the in situ data to flag and

exclude potentially erroneous measurements. We considered only errors due to systematic drift



in measurements over time (jumps or drops) and spiky measurements that are not explained by random noise. Here we used the quality control algorithms on in situ soil moisture developed by Dorigo et al. (2013) considering only stations that have at least 10 years of observations.

Because soil moisture is also coupled with runoff through the terrestrial water budget, we added an independent check for model simulations against daily river-discharge observations from the major river basins in Belgium. For this we used the inbuilt calibration feature of mHM and calibrated the model using data from river gauging stations all over the country, obtained from the Waterinfo database for Flanders (https://waterinfo.vlaanderen.be/Meetreeksen, last accessed March 2025) and the hydrometric network of discharge in Wallonia (https://hydrometrie.wallonie.be/home/observations/debit.html?, last accessed May 2025). In total we used 91 gauging stations during the calibration period (2000–2023) and 155 stations to validate the model from 1970–1999.

## 2.3 Characterizing soil moisture droughts

To characterize soil moisture droughts, we use a monthly soil moisture index (SMI), following Samaniego et al. (2013), considering the total soil water content of the root zone up to a depth of 0.5 m (We limit our analysis to this depth since groundwater in some regions is shallower than 0.5m). For each month, grid cell soil moisture is expressed as a percentile relative to that month's historical soil moisture and scaled to a range between 0 and 1.

The computation of SMI in this study is based on the methodology of Samaniego et al. (2010), which proceeds as follows. Firstly, the monthly soil moisture averaged over the root zone depth (0.5 m for this study) is extracted and used to compute a probability distribution function (PDF) $f_t(x)$ for each grid cell as;

$$f_t(x) = \frac{1}{nh} \sum_{k=1}^{n} K\left(\frac{x - x_k}{h}\right) \tag{7}$$

Where, $x$ is the soil moisture value at which the PDF is evaluated $x_1, \ldots, x_k$ represent the simulated monthly soil moisture values for month $t$ over the simulation period. Note that this conversion is done for each calendar months separately to account for inherent seasonality in SM simulations. $K$ is a Gaussian kernel function and $h$ is the bandwidth that controls the smoothness of the kernel (equation 8). The optimal value of $h$ is computed using a cross-validation criterion.

$$K(x, x_k) = \frac{1}{\sqrt{2\pi h^2}} \exp\left(\frac{(x - x_k)^2}{2h^2}\right) \tag{8}$$

The monthly grid cell SMI is then derived by integrating $f_t(x)$ and the resulting SMI values are classified into percentiles. Drought-affected grid cells are identified using a threshold





percentile $\tau$, which is commonly set at 0.2 (e.g., Svoboda et al. (2002); Samaniego et al. (2013,

2018)). This means that for a given month, a grid cell is experiencing drought if the soil mois-

ture value falls below the $20^{th}$ percentile of values for that month. According to Svoboda et al.

(2002), this percentile represents the threshold at which the magnitude of drought begins to

damage crops, cause water shortages and present high risks of fire. Next, adjacent cells where

SMI $\leq \tau$ (henceforth denoted as $SMI_\tau$) at each timestep are consolidated to form drought clus-

ters, which are defined by a minimum threshold area. Spatial clusters which share a minimum

overlapping area at consecutive time steps are then joined to form multi-temporal clusters,

each with a unique identity. For each cluster, the mean duration (months), areal extent from the

onset to termination, and the total drought magnitude, which is the spatiotemporal integral of

$SMI_\tau$ over the area affected, are computed. Following Samaniego et al. (2013), the magnitude

of each event is computed as the space-time integral of the drought duration in months over

the area under drought. This is represented mathematically as;

$$\text{TDM} = \sum_{t=t_0}^{t_1} \int_{A_t} [\tau - SMI_i(t)]_+ \tag{9}$$

$t_0$ and $t_1$ represent the onset and termination month of a multi-temporal drought event, $A_t$

is the area under drought at timestep $t$ expressed as a percent of the total domain area, and +

means the magnitude is computed only for the positive part of the function. To avoid detecting

small, isolated and short-lived dry spells as droughts, we specified a minimum threshold area

of 640 square kilometres (about 2% of total domain area) based on Samaniego et al. (2013) for

an event to be considered as a drought, and an overlap area of the same size for two drought

events at successive time steps to be considered as a single multi-temporal drought cluster.

## 3 Results

### 3.1 Model Performance Evaluation

#### 3.1.1 Soil Moisture Simulations

The daily standardized anomalies of mHM-simulated soil moisture evaluated against in-situ

observations from the ISMN are shown in Figure 3. Of the 48 stations where in situ data was

retrieved, 21 sites passed quality-control checks and were retained for validating the model out-

puts. The resulting comparison showed that the two datasets are highly temporally correlated,

with a mean Pearson $\bar{r}$=0.86 (back-transformed averages from the Fisher z-scale), although

the strength of the correlation varied with sensor depth and type. The correlation is lowest for

the top 50 mm of the soil profile ($\bar{r}$=0.81 for all networks) and increases to 0.86 for the profile

depths greater than 150 mm.





Even for the selected in situ sites, some still exhibited spurious spikes outside of random noise (shown by the red scatter points in Figure 3). We chose not to discard these points so as to preserve an adequate number of validation stations and to highlight the practical difficulty of obtaining perfectly reliable reference soil moisture data for validating model outputs.

Despite such outliers, the model simulations and ISMN observation showed similar temporal variability in soil wetness and dryness. The difference mainly occurred in the top 50 mm layer during very dry episodes when mHM produced more extreme negative anomalies than most sensors (Figure 3 (a-d)). This explains why the correlation between the datasets is the lowest at this depth. We attribute this divergence partly to a flooring effect of capacitive sensors which tend to plateau at very low volumetric water contents whereas the model continues to resolve further drying. For deeper layers, the intensity and duration of dryness were more consistent between both datasets. Finally, we note that the strength of the agreement is also influenced by the scale mismatch between mHM soil moisture, which represents average conditions over a grid cell, and the highly localized nature of point in situ measurements.

### 3.1.2 Streamflow Simulations

The skill of the model to represent daily simulated flow over the study domain is presented in Figure 4. For a robust evaluation of model performance, we retained only those stations that had at least 10 years of data and excluded stations whose peak flow did not exceed 10 $m^3 s^{-1}$. The statistics show the model performed very well in simulating daily flows, with a mean Nash-Sutcliffe Efficiency (NSE) of 0.62 and 80% of stations having NSE $\geq$ 0.5 in calibration. Validation statistics are comparable, with a mean NSE of 0.63 and 83% of stations exceeding an NSE of 0.50, which indicates good temporal transferability (Klemeš, 1986; D. N. Moriasi et al., 2007). Spatially, as Figure 4 shows, the model shows consistent performance across the domain. The model achieved the highest performance (NSE $\geq$ 0.75) in large basins, as the model could delineate the drainage extents of such basins with higher accuracy. This delineation becomes more challenging in smaller basins and especially where the topography is less pronounced, as is the case in the northern part of the domain. Accordingly, the lowest model performance was observed in gauging stations draining the smallest basins. In some cases, anthropogenic modification of rivers such as canalization, diversions and diking, which is common in the northern lowlands and which are not implemented in the model, explained poor model performance at some gauging stations. Notwithstanding these few cases, the results demonstrate that the model provides a reliable, spatially consistent basis for assessing soil moisture dynamics over the country.





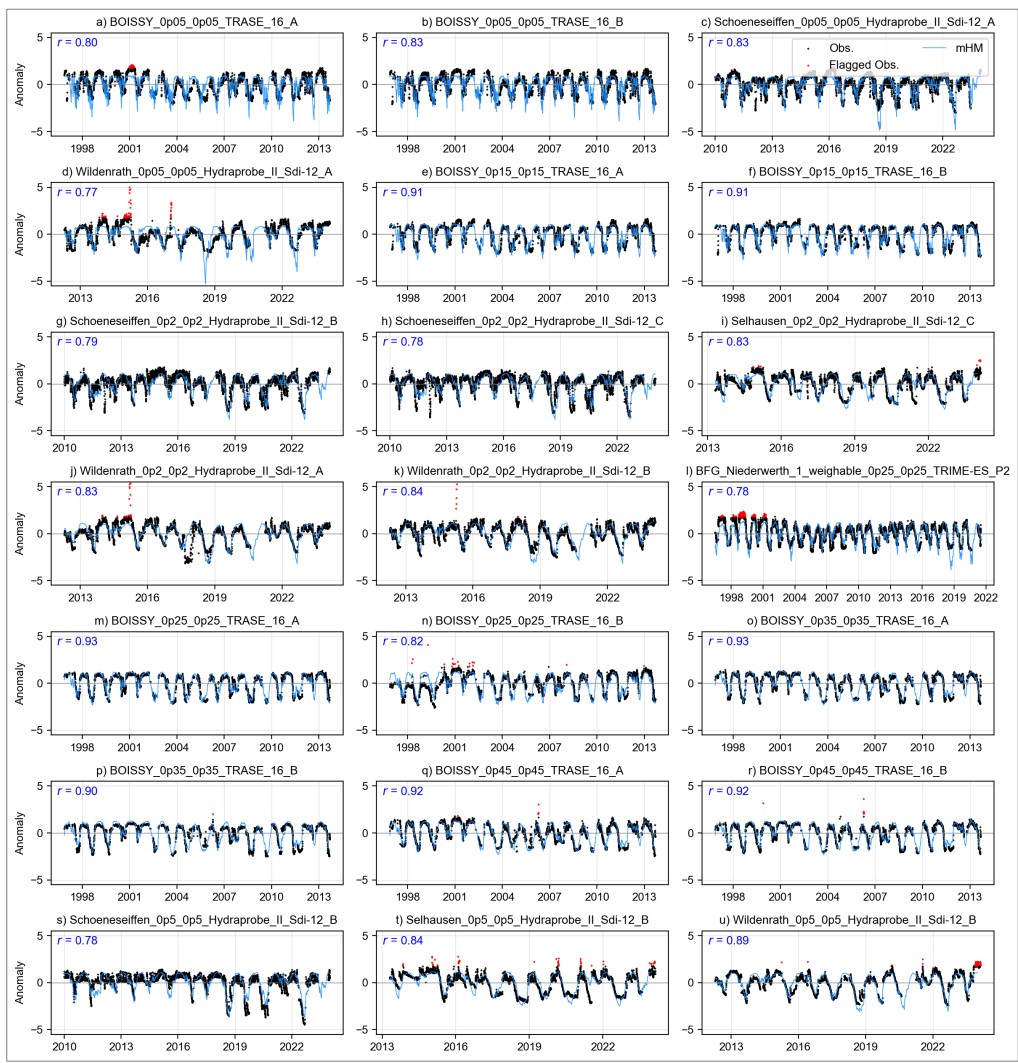

**Fig. 3**: Comparison of standardized anomalies between mHM and in situ soil moisture at selected ISMN sites, ordered by increasing sensor depth. The red scatter points represent observed soil moisture values flagged as potentially erroneous. Titles follow the format station_topdepth_bottomdepth_sensortype, e.g., BOISSY_0p05_0p05_TRASE_16_A refers to the Boissy station with a sensor at 0.05 m depth and sensor type TRASE.





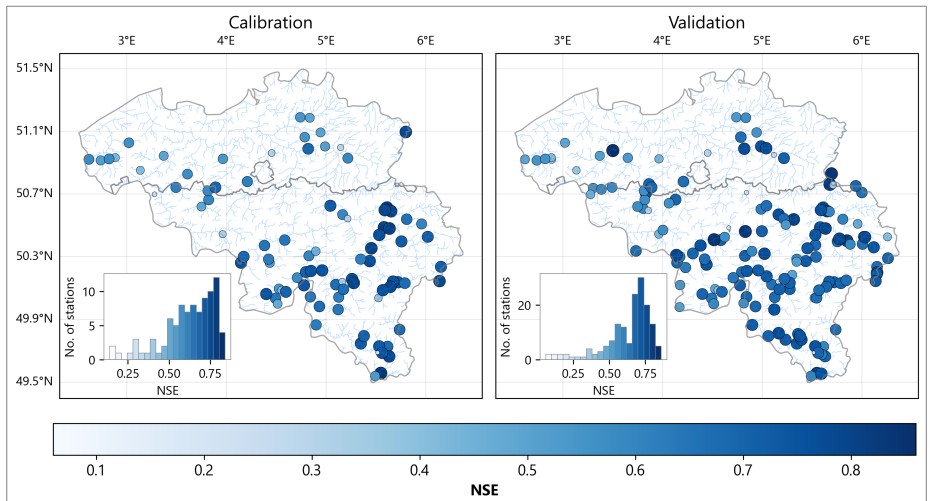

**Fig. 4**: Model performance at gauging stations across Belgium during calibration and validation periods. The colour intensity and size of each circle are proportional to the NSE value. The inset histograms show the distribution of NSE values across all stations for each period.

## 3.2 Multidecadal evolution of soil-moisture droughts

Figure 5 shows simulated soil moisture droughts over Belgium between 1970 and 2023. The events are ranked by Total Drought Magnitude (TDM), the cumulative deficit in soil moisture below the chosen drought threshold (SMI $\leq 0.20$), integrated over the area and duration of the drought event. The biggest ten events ranked by TDM are colored and annotated with their corresponding periods. From an interdecadal perspective, the figure reveals three distinct drought regimes. Three drought events are apparent in the 1970s, which are dominated by the historic 1975–1977 droughts. Although this event is commonly referred to as the 1976 drought, probably because that is when it peaked, the analysis shows that its development in Belgium began back in the autumn of 1975 and lasted for a record 16 months until the winter of 1977. By the end of the event, 63% of the domain had experienced drought conditions although this fluctuated at different times[1]. This event established a benchmark against which subsequent drought events in Europe are commonly judged against. Our analysis reflects this, as this event matches the most intense drought in Belgium in the 53 years since 1970. Henceforth, this decade will be referred to as the 1971–1980 decade (we disregard 1970 because it is a calibration period for the drought analysis).

---

[1]The 63% figure is the mean fraction of the domain affected across all time steps during the drought; at individual times coverage ranged below and above this value, with a maximum of complete (100% ) coverage when the drought peaked




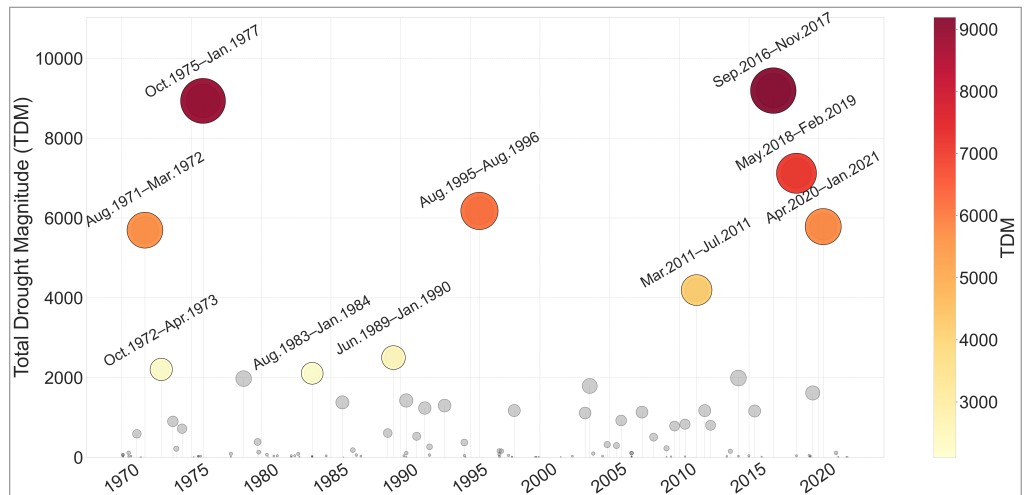

**Fig. 5**: Duration and magnitude of drought events from 1970 to 2023. Each circle represents a drought event, positioned according to its start date (x-axis). The circle size is proportional to the Total Drought Magnitude (TDM) of each event. The ten most severe droughts, ranked by TDM, are highlighted with coloured markers, with their corresponding periods annotated.

A relatively wetter hydroclimatic regime characterizes the three subsequent decades (1981-1990, 1991–2000 and 2001–2010). This is observable in Figure 5 as indicated by lower-magnitude drought occurrences. In these three decades, only three drought events are big enough to feature in the top ten droughts, and even these ranked relatively low in the TDM scale. The 1995-96 drought, the biggest of the three, however did persist for at least a year. A significant shift in drought frequency and severity emerged after 2011. Of the ten highest ranking droughts from 1971, 40% of them were recorded in the 2011–2020 decade with three severe drought events clustered in rapid succession between 2016 and 2020. The 2016–2017 drought is the biggest in this decade, matching the 1975–1977 drought by magnitude, affected area (64%) and lasting nearly as long (15 months) before it fully dissipated. The 2018-19 droughts also rank highly although it lasted about 10 months but affected a bigger area on average (73%). This pattern has continued robustly into the 2020s, as underscored by the 2020–21 and 2022–2023 droughts. By cumulative magnitude, the 2022–2023 event, not shown here, (lasted for 12 months between March 2022 and February 2023 with a TDM of 7870) ranks just below the 1975–1977 and 2016–2017 droughts. We have excluded drought events after





2020 from the subsequent decadal analysis because the current decade is still incomplete. In
the subsequent analysis, the 2020–2021 is also only considered until the end of 2020.

## 3.3 Area characteristics and shifts in drought-class composition

Figure 6 combines the temporal and areal characteristics of drought to illustrate the proportion
of the domain experiencing varying degrees of soil-moisture drought severity through time.
The categorization of drought into severity classes is based on Svoboda et al. (2002). The
categories are clustered as follows; moderate drought ($0.1 < \text{SMI} \leq 0.2$), severe drought ($0.05
< \text{SMI} \leq 0.1$), extreme drought ($0.02 < \text{SMI} \leq 0.05$) and exceptional drought ($\text{SMI} \leq 0.02$).
For conciseness we will examine the changes at both ends of the drought spectrum.

During the 1971-80 decade, droughts were predominantly moderate ($0.1 < \text{SMI} \leq 0.2$).
When they did occur, exceptional droughts did not affect more than 30% of the domain at
their peaks in 1971–1972 and 1976-1977. The figure also shows that these two droughts were
disrupted by wetter spells which allowed re-establishment of normal to wet soil moisture con-
ditions. When accumulated over the decade, moderate droughts accounted for about 75% of
all grid-cell months affected by drought, while exceptional droughts, which are very rare by
design, accounted for about 3% of drought-affected area, most of this occurring during the
1975–1977 drought (donut plots, Figure 6).

Normal to wet conditions interspersed with episodic, short-lived droughts dominate the
spatiotemporal profile between 1981 and 2010. Decadal accumulations show that at least 80%
of all drought occurrences during this time were moderate in intensity, while exceptional
droughts constituted, on average, less than 1% of occurrences over the three decades.

In contrast, the 2011–2020 decade experienced more frequent and severe droughts, particu-
larly towards the end of the decade. In comparison to the previous decades, the spatial footprint
of exceptional droughts noticeably increased. At the peak of the 2011 and 2016–2017 droughts,
more than 40% of the drought-affected area was under exceptional drought, which did not
previously occur even during the 1975–1977 event. This increase is reflected in the decadal
drought area severity, where exceptional droughts accounted for 5.9% of drought-affected area,
exceeding all the previous four decades combined.

## 3.4 Decadal drought exposure

Complementing the temporal and spatial analyses, Figure 8 illustrates decadal drought persis-
tence, expressed as the total number of months in which each grid cell experienced $\text{SMI} \leq
0.2$ in a decade. The results agree with those of the previous analysis. During 1971–1980, the
domain accumulated between 12 and 36 drought months, with a domain-wide mean of about
24 months per grid cell (2.4 months/year), (inset histogram).





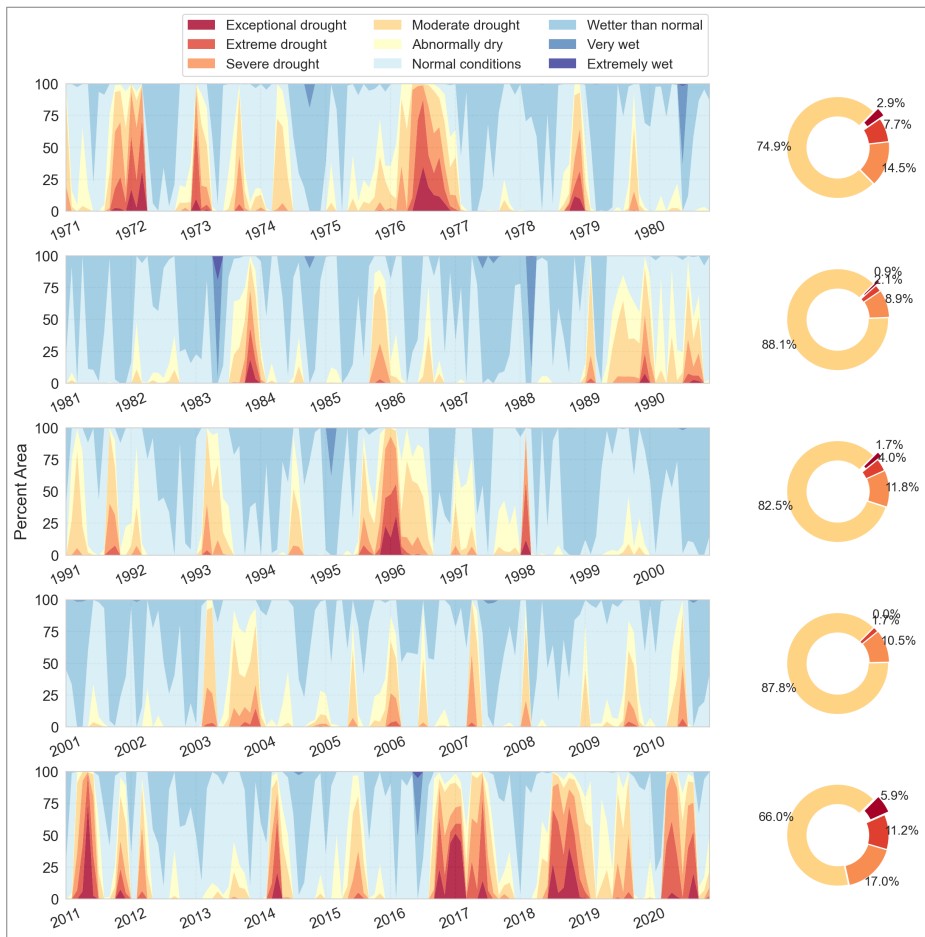

**Fig. 6**: Decadal spatio-temporal evolution of soil moisture in Belgium, 1971–2020. For each decade (stacked panels, left) the coloured bands show the percentage of land area falling into soil-moisture classes, from "exceptional drought" to "extremely wet". The accompanying donut charts (right) aggregate only the months in which some part of the country was in drought (SMI ≤ 0.20); they display the mean share of the drought-affected area that fell into each drought class over the decade. Months without drought contribute no area to the donut.

Domain-wide improvements in moisture conditions are apparent in the next three decades. The mean cumulative totals fell to 13 months in 1981–1990 (1.3 months/yr), 17 months in 1991–2000 (1.7 months/yr.), and 14 months in 2001–2010 (1.4 months/yr).





As with the other metrics, drought persistence peaked in 2011–2020. The domain accumulated between 24 and 48 months of drought over the decade, and the domain-wide mean rose to 37 months, or 3.7 months per year (Figure 8). To put this into perspective, this amounts to roughly three continuous years of soil-moisture drought within the decade. This cumulative exposure is more than twice that of each of the three preceding decades (1981–1990, 1991–2000, 2001–2010) and about 1.5 times higher than the previous driest decade 1971–1980.

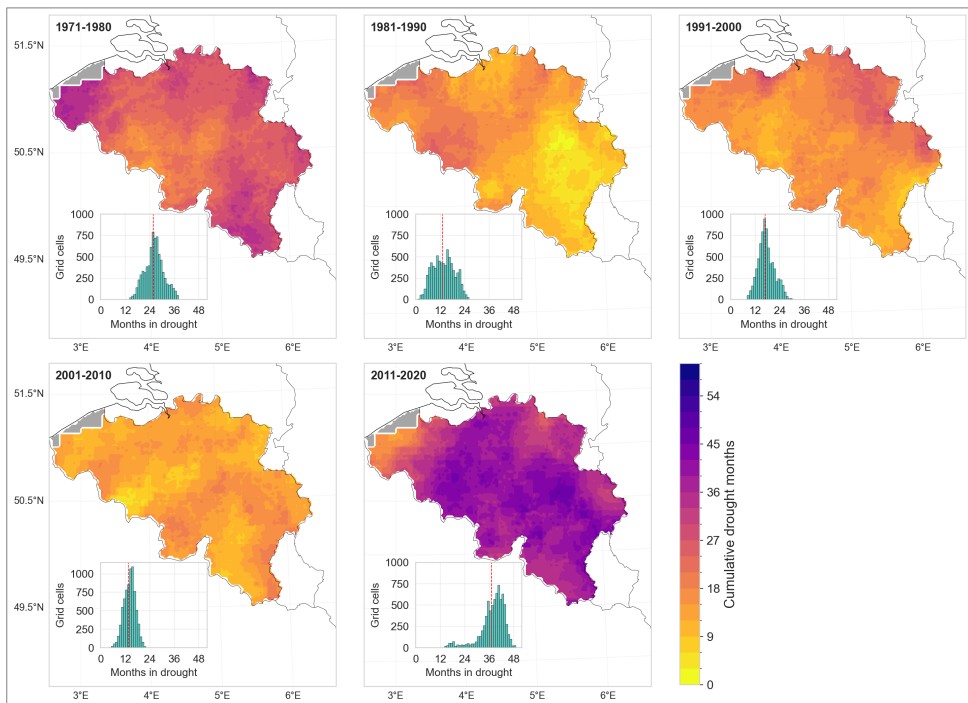

**Fig. 7**: Distribution of the number of months within each decade that a grid cell experienced drought conditions (SMI ≤ 0.2). The inset histograms show the frequency distribution of cumulative time under drought for all grid cells. The red dashed line indicates the mean duration. EOBS data is missing for the region shaded grey.

To test whether 2011–2020 was statistically drier than the preceding four decades, we applied a non-parametric bootstrap to the per-pixel cumulative drought durations (SMI ≤ 0.20) and to the subset of exceptional drought months (SMI ≤ 0.02). For each decade, we generated 100,000 bootstrap samples by resampling grid-cell drought durations with replacement,



calculated the mean for each sample, and used the 2.5$^{th}$ and 97.5$^{th}$ percentiles of the resulting distribution to derive the 95% confidence interval (CI) of the sample mean.

The statistical analysis concludes that 2011–2020 was indeed the driest decade of the five decades, both in terms of total drought duration and exposure to exceptional droughts. Over the decade, Belgium accumulated a mean drought period of 37 months (CI:36.9–37.2 months), significantly higher than in 1971–1980 (mean=25.65 months [CI: 25.6–25.8]), which is the next driest decade (Figure **??** (a)). The lower bound of the 2011–2020 decade CI lies 11 months above the upper bound of the 1971–1980 period and far higher than those experienced in the three decades in between (1981–1990: mean 13 months [CI: 12.92–13.15], 1991–2000: mean 16.9 months [CI: 16.80–16.95] and, 2001–2010: mean 13.52 months [CI: 13.46–13.59]).

A similar contrast emerges for the most severe drought (Figure **??**(b)). The 2011–2020 decade accumulated 4.3 months of exceptional drought on average (CI: 4.28–4.38), more than the combined total of the four earlier decades. None of the previous decades reached a mean of 2 months of exceptional droughts. 1971–1980 accumulated 1.94 months (CI: 1.89–1.98), 1981–1990 only 0.35 months (CI: 0.34–0.36), 1991–2000 0.80 months (CI: 0.79–0.84), and 2001–2010 experienced virtually no exceptional drought. In cumulative terms, more than half of all exceptional drought months in the five-decade record occurred between 2011 and 2020.

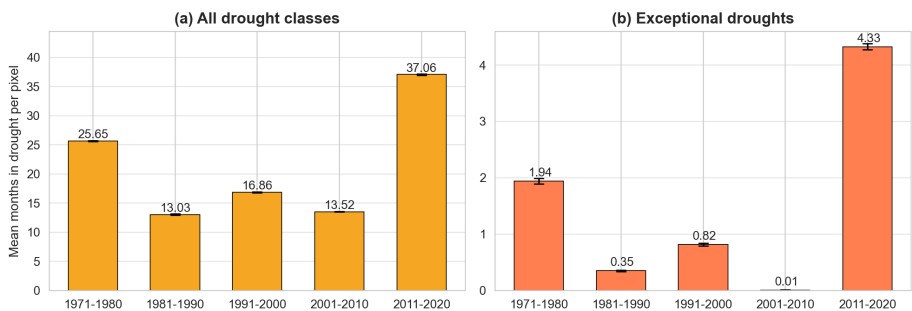

**Fig. 8**: Decadal pixel-wise drought persistence. The bars show the mean number of months each grid cell spent in drought per decade, with 95% bootstrap confidence intervals (black whiskers) for (a) All drought classes (SMI ≤ 0.20) and, (b) Exceptional drought only (SMI ≤ 0.02).





## 3.5 Divergence Between Soil Moisture and Precipitation-Based Drought Indicators

To investigate how precipitation-based drought indicators reflect land surface moisture stress, we compared the SMI and SPEI during the most severe soil moisture drought events ranked by TDM (1975–1977, 2016–2017, and 2018–2019). Since the SMI is computed on a monthly timescale, we calculated the accumulated difference between EOBS precipitation and potential evapotranspiration at one- and three-month timescales and used the SPEI package developed by Vonk (2024) to compute pixel-wise 1-month SPEI (SPEI-1) and three-month SPEI (SPEI-3). We also limit the accumulation period to 3 months as this is what is currently used in drought monitoring in the country. As SPEI is an anomaly-based rather than a percentile-based index, we associated an SPEI value of -1.0 to an SMI value of 0.2 to represent the threshold for moderate drought severity, according to the guidelines by Svoboda et al. (2002).

Across the three drought events, SMI generally exhibits more persistent negative anomalies than SPEI-1 and, to a lesser extent, SPEI-3 (Figure 9). SPEI-1 is highly responsive to short-lived rainfall deficits and surpluses that may not immediately alter root-zone storage; this sensitivity captures meteorological conditions which differ from soil moisture conditions that integrate past deficits through slow infiltration and plant uptake. As expected, SPEI-3 smooths some of the short-term variability inherent to SPEI-1 and more closely mirrors the temporal pattern of soil-moisture anomalies. Even so, for our domain, SPEI-3 still tends to underestimate the persistence and the magnitude of deficits relative to SMI. For example, of the three drought events, SMI shows that soil moisture anomalies were strongest during the 2016–2017 drought (SMI near zero). SPEI-3 on the other hand appears to underestimate the extent of this difference between the three droughts. SMI also shows a stronger persistence in time, which implies that soil moisture has a higher inertia and responds not only to the magnitude but also to the sequence of meteorological anomalies.

When analyzing drought recovery, the same pattern also emerges. SPEI-1 reacts fastest and shows an earlier termination of droughts. Although the exact pattern of recovery is event-specific, drought recovery follows the same general order; short-term water balance anomalies (SPEI-1) normalize first followed by seasonal water balance anomalies (SPEI-3) before soil moisture conditions emerge out of drought. This pattern is most evident during the drought events of 1975-1977 and 2018–2019 (Figure 10). During the 1975–1977 drought event, all the indices show that the drought-affected area peaked by August 1976. According to the evolution of SPEI-1, the drought had virtually terminated by around November 1976. Yet, by this time almost half of the domain area was still under SPEI-3 drought while SMI shows closer to 90% of the domain was still under drought. By the time SPEI-3 drought terminates in January 1977, more than one-third of the domain was still under SMI drought, which took until February





1977 to terminate. A similar sequence of recovery is observed during the 2018-2019 drought. The 2016–2017 drought was interrupted by intermediate wet conditions during March and April 2017 which led to partial drought recovery and consequently a smaller margin between SPEI-3 and SMI recoveries.

We stress that these differences do not imply that one indicator is necessarily *better*; rather, they are all useful for demonstrating how a drought shock progressively propagates through different components of the hydrological system. Precipitation-based indices like SPEI reflect short-term meteorological inputs that may still be agriculturally meaningful. As Figure 9 shows, rainfall events during dry summers may not replenish deeper soil moisture due to immediate losses through evapotranspiration, yet these events can still temporarily alleviate plant water stress, especially for fast-responding, shallow-rooted crops or annual crops. The recovery of SPEI out of drought conditions may thus signal 'relief' that is real, albeit short-lived and limited in scope. SMI-based drought analysis better captures the persistence of land surface water deficits and the residual moisture stresses that continue to affect the dependent ecosystems (e.g. perennial deep-rooted vegetation) long after meteorological conditions have normalized.

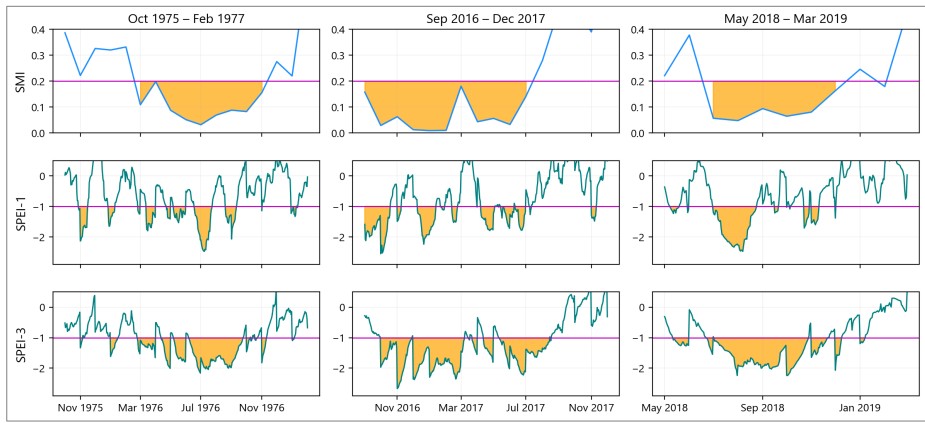

**Fig. 9**: Comparison of domain-average SMI, SPEI-1, and SPEI-3 time series during the three biggest drought events up to 2020. The orange shaded areas indicate drought conditions, defined as SMI $\leq$ 0.2 and SPEI $\leq$ -1.0. The horizontal magenta lines mark the drought threshold for each index.





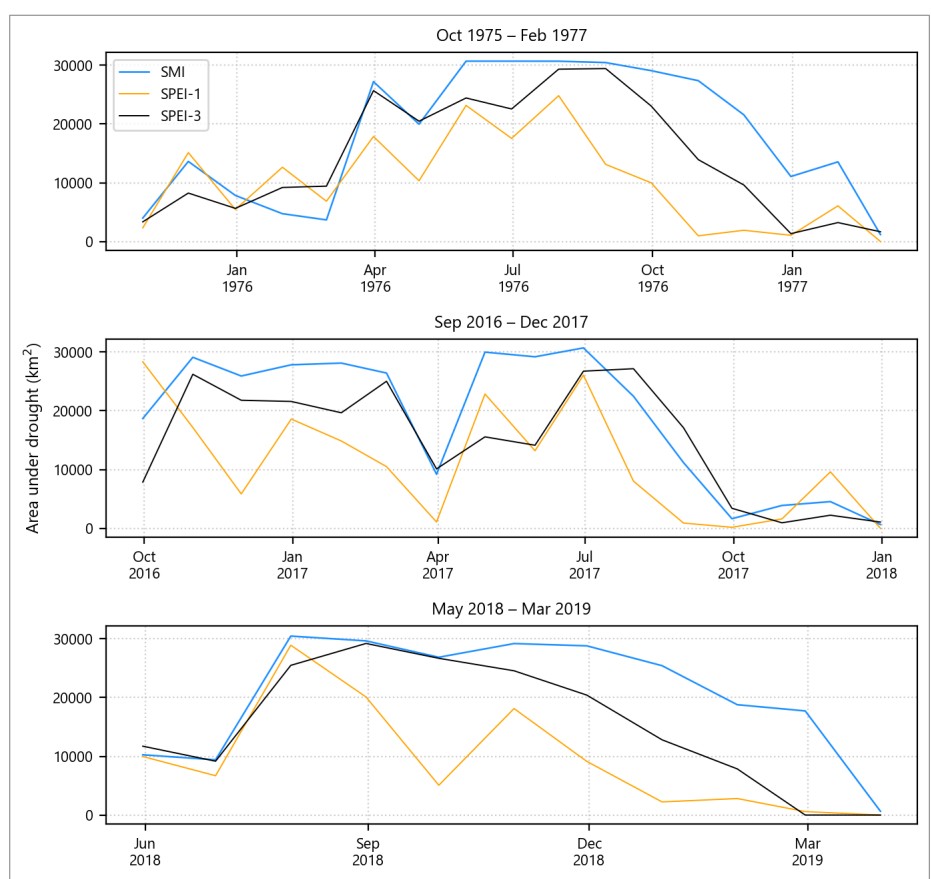

**Fig. 10**: Evolution of area affected by drought during the three biggest drought events, represented using the three indices. The thresholds for SPEI and SMI are as described in Figure 9

.

## 4 Discussion

This extended temporal analysis of soil moisture droughts over Belgium offers new insights on the severity of recent droughts in the country. Without such a long-term, multi-decadal viewpoint, the recent intensification of drought severity and frequency might be mistakenly viewed as isolated, transient events rather than as indicators of a potential shift in the climate regime. These changes, despite the absence of significant linear trends, also raise important questions regarding potential non-linear transitions in regional hydro-climatic equilibria, to which we find answers by studying longer reconstructions of the European drought patterns from other studies.





Our findings are consistent with the wider pan-European narrative of intensifying droughts over the continent in the $21^{st}$ century. García-Herrera et al. (2019) showed that drought conditions covered 90% of central-western Europe from July 2016 to June 2017, with 25% of the area in record-breaking severity. This drought led to widespread impacts of agriculture, water supplies and hydropower production and was the most severe drought Europe had faced between 1979 and 2017. Longer historical reconstructions of European droughts by Hari et al. (2020) and Rakovec et al. (2022) show that the occurrence of the consecutive European summer droughts of 2018–2019, where 50% of Central Europe was under extreme drought conditions, is unprecedented in the last 250 years (since at least 1766). In their synthesis of the effect of this drought on crop yields, the same study found that the drought reduced maize yields in western Europe by 20-40% and caused about a 10% loss in barley yields for a majority of European countries. By dating stable tree-ring isotopes to reconstruct the summer hydroclimate of central Europe from 75 BCE to 2018 CE, Büntgen et al. (2021) found that the recent succession of extreme European summer droughts between 2015 and 2018 are unprecedented in the previous 2,110 years.

Studies attribute atmospheric circulation patterns and the potential role of anthropogenic warming as the dominant drivers of these drought dynamics. Ionita et al. (2020) link the sustained period of spring droughts in Europe between 2007 and 2020 to a prevalence of anticyclonic and a persistent blocking high over the North Sea. These conditions deflect westerly storms and increase temperature due to a lengthened sunshine duration. This consequently increases evapotranspiration, which has been found to amplify European summer droughts (Teuling et al., 2013). García-Herrera et al. (2019) observed that high-latitude atmospheric blocking contributed to the drought conditions over northwestern Europe in 2016–2017 by decreasing moisture transport from the Atlantic Ocean. Hari et al. (2020) similarly attributed the intensification of the 2018-2019 drought to anticyclonic circulation which caused a blocking that increased temperature anomalies to +2.8 K in central to northern Europe (Rakovec et al., 2022). These patterns are projected to persist in future as anthropogenic warming weakens the temperature gradient between the polar and mid-latitudinal regions and fluctuate the strength of the jet stream and the persistence of extreme weather events (Cohen et al., 2014; Dai et al., 2019; Ionita et al., 2020). Europe-wide studies show that anthropogenic warming will worsen droughts and events with the nature of severity as the 2018–2019 drought will become routine, persist longer and affect a larger proportion of area (Samaniego et al., 2018; Hari et al., 2020; Rakovec et al., 2022). These emphasize the need to continue strengthening drought monitoring and investing in drought preparedness and mitigation measures.

On the comparison between precipitation and soil-based drought indicators, we stress that these indicators are useful for different components of the hydrological system. SPEI-1 and





SPEI-3 may suit analyzing drought patterns in shallow soil layers and shorter temporal scales but are limited for indicating drought persistence deeper in the soil or in complex ecosystems due to their ignorance of land-ecosystem interactions (Xu et al., 2021; Peng et al., 2024). When assessing drought impacts on ecosystems, groundwater recharge, or perennial vegetation like forests, the divergence between meteorological and soil moisture signals can become complex. In such systems, soil properties such as hydrophobicity during prolonged dry periods can lead to highly uneven infiltration (Gimbel et al., 2016; Filipović et al., 2018). Heavy summer rainfall may not be absorbed uniformly across the soil profile, but instead run off or infiltrate preferentially along cracks, roots, or macropores, sometimes bypassing the upper root zone. While this limits the ability of standard soil moisture indices to reflect actual water availability near the surface, it may still benefit deep-rooted vegetation like trees by replenishing deeper soil layers (Zhu et al., 2015; Duniway et al., 2018). Assessing drought stress and recovery in these systems thus requires models and indicators that account for vertical and spatial heterogeneity in infiltration and root water uptake (e.g., Shen et al. (2025)), rather than relying solely on averaged or surface-weighted soil moisture metrics. Further, while it may be argued that SPEI at longer accumulation periods (e.g., 6, 9 or 12 months) can lead to a closer resemblance of root zone moisture conditions, finding the appropriate accumulation lengths is dependent on landscape and soil characteristics (topography, rooting depth, soil hydrology and management conditions) and climatic conditions, which can lead to a strong variation of drought characteristics if the landscape is heterogeneous. Kumar et al. (2016) indeed found that applying spatially variable accumulation periods achieves a higher correlation between precipitation-based and groundwater drought indices, over a uniform domain-wide accumulation period, even at long accumulation times.

## 5 Limitations and future work.

Our results rely on the evaluation of model-derived soil moisture conditions, which are inevitably constrained by structural, parametric, and forcing uncertainties that we did not explicitly evaluate. Choices of the mapping between drought categories (e.g., SPEI $= -1.0$ vs. SMI $\leq 0.2$) and a uniform accumulation period over the whole domain (for SPEI analysis) also introduce additional subjectivity. mHM model does not also account for anthropogenic factors such as irrigation, groundwater abstraction, tile drainage and artificial canals, and land management conditions, which affect the hydrology of the domain. Future work can partially offset these limitations by quantifying uncertainty using ensembles of forcings, investigating model parameters to derive confidence intervals for drought magnitude, area, and timing, incorporating human water use and irrigation processes, or assimilating independent observations



(such as in situ or remotely sensed soil moisture and terrestrial water storage) to better constrain states and evaluate the joint behaviour of multiple drought indicators alongside observed impacts.

## 6 Conclusion

Our multi-decadal, high-resolution analysis of rootzone soil moisture dynamics over Belgium reveals that soil moisture droughts experienced in the country during the 2011-2020 decade were the worst the country had experienced since at least 1971. Our analysis shows that droughts in 2011-2020 occurred nearly twice as frequent compared to the preceding three decades and exceeded even the historically severe droughts of the 1970s in both duration and intensity. By studying recent patterns in droughts over Europe, we found that this pattern is part of a broader, continent-scale shift toward more persistent droughts. Studies show that the recent rapid succession and increased severity of droughts in the latter part of the 2010s is unprecedented even in millennial timescales, an indicator that these anomalies might not be occurring within a stationary climatic regime. These could rather be signals of a transition towards conditions where droughts become longer and recur more frequently, driven by large-scale atmospheric blocking events that favour the persistence of higher temperatures that enhance evapotranspiration.

This study also shows that characterizing agricultural droughts using indices based on soil moisture offers a more holistic representation of land surface water stress compared to precipitation-based drought indices. While current drought assessments in Belgium rely upon meteorological indices (SPI and SPEI), this study shows that these indices can underestimate the persistence and severity of soil moisture drought conditions in the root zone, which often lag meteorological recovery, due to the memory effect of the land surface. Including soil moisture monitoring in drought observatories thus offers the added value of capturing lingering stresses on agriculture and cosystems, which can persist long after meteorological conditions have normalized. This gives decision-makers a better view of drought severity and duration and guides them on how to devise the appropriate response and mitigation efforts. Lastly, as anthropogenic warming worsens the occurrence of droughts, recognizing and proactively planning for this evolving drought paradigm will be crucial for ensuring the resilience of water-resource management, agriculture, and ecosystems in a warming climate.



**Author Contributions:**

KL, RK and OR formulated the study and set up the model simulations. KL analyzed the data and prepared the figures with contributions from OR, RK and SD. All authors contributed to writing and reviewing the contents of the manuscript. All authors read and approved the contents of the final manuscript.

**Acknowledgements:**

We acknowledge the High Performance Computing of Vrije Universiteit Brussel for providing the computational resources required to run the model and the analysis of model outputs. We also acknowledge all the sources of data used in this study for providing the data openly.

**Funding:**

The authors acknowledge the financial support of the Research Foundation – Flanders (FWO) for funding the International Coordination Action (ICA) "Open Water Network: Impacts of Global Change on Water Quality" (project code G0ADS24N). OR acknowledges the Research Excellence in Environmental Sciences (REES) project of the Faculty of Environmental Sciences, Czech University of Life Sciences Prague.

**Data Availability:**

All datasets used in this paper are openly available as described in the methodology text.

**Code Availability:**

The scripts used to arrive at the findings of this study is available at:

https://github.com/klekarkar/mHM_IO_dataprocessing.

The SMI analysis was carried out using the SMI package, available at:

https://github.com/mhm-ufz/SMI.

**Competing interests:**

At least one of the (co-)authors is a member of the editorial board of Hydrology and Earth System Sciences.



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
