# Peer review of "Soil moisture droughts in Belgium during 2011–2020 were the worst in five decades"

_EGUsphere, 2025_

## Author Comment (AC3)

**Author Responses to Reviewer #2 Comments**

Katoria Lekarkar*

We would like to begin by thanking the reviewer for taking the time to review our manuscript and provide their comments. These will go a long way to improving the manuscript. In the sections that follow, we provide our responses to individual reviewer comments on the manuscript. We do note, however, that in some cases, the questions/suggestions raised by the reviewer are already addressed by the manuscript. In these cases, we will point out which sections of the manuscript already address the reviewer comments. The responses to individual comments are written in this **color**.

**1: Overall assessment**

**Reviewer comment:**

This study analyzes the root-zone soil moisture dynamics in Belgium from 1970 to 2020, focusing on the severity and persistence of droughts during the 2011–2020 period. It highlights the unprecedented nature of these droughts and evaluates the limitations of precipitation-based drought indices (such as SPEI) in drought assessments. The paper proposes using root-zone soil moisture as a more effective drought monitoring indicator and underscores the increasing frequency and persistence of drought events in the context of climate change. The work is valuable and substantial, but its scientific novelty is limited, and some methodological and analytical aspects need further clarification or strengthening.

Specifically, the manuscript currently lacks critical quantitative evidence to support two core claims: (1) that root-zone soil moisture provides added operational value over precipitation-based indices (SPEI), and (2) that the chosen reconstruction approach (mHM) is preferable to widely used soil-moisture products (e.g., ERA5-Land). I recommend the authors (a) perform an event-based contingency analysis comparing mHM-RZSM and SPEI drought events (onset, termination, duration, severity) and report hit/miss/false-alarm statistics, and (b) provide a direct inter-comparison with at least one widely used soil-moisture product to quantify time-series agreement and event-detection differences.
* * *
*Corresponding author: katoria.lesaalon.lekarkar@vub.be

**Author response:**

The manuscript presents the first high-resolution, multi-decadal reconstruction of root-zone soil moisture droughts for Belgium, where long-term in situ soil-moisture observations or any studies of the kind we did are not currently available. The primary scientific contribution lies in combining this reconstruction with a spatially explicit drought-event analysis and a mechanistic comparison between precipitation-based and soil-moisture-based drought indicators. This combined approach allowed us to determine how the droughts experienced in recent years compared with those experienced in the last 50 years and enabled us to understand their rarity. No study of this kind has been done in Belgium.

We agree that distinguishing drought types and explaining the conceptual advantages of root-zone soil moisture is important. We would like to point out that the current manuscript already includes a detailed discussion of these points (Lines 88–103), where we explicitly describe the limitations of precipitation and temperature-based indices such as SPEI for capturing agricultural drought conditions. In the manuscript we report that the precipitation-based indices are not able to represent the vertical distribution of water in the root zone or vegetation water stress, the nonlinear and lagged response of soil moisture to precipitation, and the soil-moisture memory effect that often leads to prolonged drought persistence. We feel that these points reinforce the added operational value that soil moisture provides. As for the quantitative evidence, we address this in Section 3.5 of the manuscript, where we show that: (i) SPEI-based droughts terminate earlier than soil moisture droughts and (ii) SPEI-based droughts underestimate the severity of droughts compared to soil moisture droughts. To add more depth to this, we will compare other metrics like the number of months under drought conditions during the major drought events we discuss therein.

On the merits of using mHM over other soil moisture products, we used mHM in our study to generate soil-moisture fields at high spatial resolution. We generated soil moisture fields at $1/32^0$, which is a resolution high enough to account for the heterogeneities that characterize land surface conditions that affect soil moisture. This higher resolution is better suited for drought analysis since it takes into account the heterogeneity in soils, land use and land cover all of which vary strongly at short distances. ERA5-Land is indeed a valuable reanalysis product but it is better suited to for large-scale applications (continental to global scales). The $\approx 9$ km resolution of ERA5-Land limits its suitability for local-scale drought monitoring as the resolution generalizes land surface conditions. Due to this generalization of land surface heterogeneity, coarse-scale products like ERA5-Land can produce biases in surface water and energy fluxes (Crow et al., 1999). To illustrate, if we used the 9 km ERA5-Land over Belgium (about 30,000 sq. km in area), we would obtain about 370 grid cells. On the other hand, our model resolution generates 3,100 grid cells, almost ten times more than ERA-5 Land. Our model resolution is thus better at differentiating fine-scale heterogeneities in soil moisture conditions and deriving local-scale variability.

One would also ask why we did not use remote sensing (RS) soil moisture products. While

RS offers an alternative source of soil moisture, RS-derived soil moisture only measures water content in upper few centimetres of the soil profile and have low quality under certain surface conditions such as dense vegetation, frozen soils and mountainous terrain. The data can also often be missing due to retrieval conditions or satellite revisit times (Wang et al., 2011; Peng et al., 2017).

**2:   Major comment #1**

**Reviewer comment:**

Please clarify the definition of the root-zone layer used in this study (0–0.5 m). Why was this depth chosen, and how representative is it across different vegetation types and land-cover conditions in Belgium? Considering the variability of underlying surfaces could help assess the robustness of the results.

**Author response:**

In the manuscript we have clarified that the root-zone depth was limited to 0–0.5 m because in most areas of the country, groundwater occurs below this depth (Lines 263-264), thus including deeper layers would risk mixing soil-moisture dynamics with groundwater storage. We agree that representativeness across vegetation types can vary. For clarity, We will add a sentence justifying that this depth corresponds to the dominant rooting depth of shallow-rooted croplands, grasslands, and some vegetation types that are disproportionally sensitive to moisture availability in the unsaturated zone and thus more susceptible to soil moisture droughts.

**3:   Major comment #2**

**Reviewer comment:**

The Introduction successfully establishes the severity of drought in Belgium and correctly identifies the scientific gap regarding the limitations of precipitation-based indices. However, the section's structure and balance require revision to maximize clarity and scientific impact.

- The lengthy, detail-heavy descriptions of the 2011, 2018–2019, and 2022 droughts (Lines 46–74) read like an event chronicle. This narrative must be significantly condensed to focus only on the key messages that motivate the need for a long-term assessment, ensuring the scientific gap is presented more prominently.

- Conceptual Distinctions: The discussion on drought monitoring (Lines 79–103) should be enhanced by more explicitly distinguishing meteorological/hydrological drought from agricultural drought. This includes clarifying why traditional indices like SPEI are limited and emphasizing the superior conceptual role of root-zone soil moisture (RZSM) for capturing plant water stress.

- The final paragraph (Lines 104–116) prematurely introduces technical specifics (e.g., mHM model, offline forcings, SMI derivation via percentile ranking). These details should be relocated entirely to the dedicated Methods section.

**Author response:**

We do agree that the introduction can be improved in terms of structure and balance. As suggested by the reviewer, we will make the description of the drought events more concise without losing important details. On the second point, we would like to point out that the conceptual distinction between meteorological and agricultural drought is already addressed in Lines 84–103, where we explicitly describe the limitations of precipitation and temperature-based indices such as SPEI for capturing agricultural drought conditions. In the manuscript we report that the following limitations of precipitation-based indices:

- They do not represent the vertical distribution of water in the root zone or vegetation water stress

- The response of soil moisture to rainfall is lagged and non-linear, which is not reflected by precipitation-based indices

- They do not account for soil-moisture memory effect that often leads to prolonged drought persistence

We feel that these points reinforce the added operational value that soil moisture provides. To improve clarity, we will add an explicit definition that differentiates the drought types more succinctly. On the final paragraph, we do agree that there are some details that can be moved to the methodology section. We will thus modify it to mainly focus on the objectives of the study with less emphasis on the methods we used.

**4: Major comment #3**

**Reviewer comment:**

The authors provide a compelling and well-referenced justification for prioritizing the Pearson correlation coefficient to assess the temporal agreement of standardized soil moisture anomalies. However, two critical components are missing for a complete validation of the model's skill in the context of this drought study. To fully characterize the model's performance beyond just temporal consistency, the authors should consider reporting an appropriate error metric, such as the Unbiased Root Mean Square Error (ubRMSE). The ubRMSE is ideally suited for this validation context, as it quantifies the error component related to the model's random fluctuations and timing errors, while excluding the systematic absolute bias that is deliberately factored out by the standardization approach. Crucially, given that the study's goal is drought analysis, the evaluation should include an explicit assessment of the model's

ability to accurately represent drought conditions as observed by the in situ data. For instance, comparing the model's and in situ data's ability to correctly classify dry/drought days based on an established threshold (e.g., the 20th percentile), assessing the correlation or error between the model-derived soil moisture index (SMI) and an index derived from the in-situ data.

**Author response:**

Thank you for this comment. The current analysis focuses on correlation because the soil moisture time series are standardized, and correlation directly assesses temporal agreement and anomaly timing, which is what we are interested in from a drought analysis perspective. As the data comparison already shows, there is a high agreement between the two datasets. We feel that adding more statistics would lengthen this section and given that it is not the main story of the article, it would risk dragging the attention of the reader away from the main message. Reviewer #1 already commented that we have used quite a number of statistical metrics in the manuscript. Adding more of them will thus be added confusion to the reader.

On the use of the SMI for comparing the two datasets; since the in situ data has missing values in some stations, it would be difficult to do the SMI analysis since this requires aggregating daily values to monthly averages. This would thus introduce errors in the analysis if we include months that have days of missing data. The reviewer has an important suggestion though that we can do a percentile-based analysis for comparison. Together with that, we can add the uRMSE but we suggest to make these changes as minimal as possible so, preferably as a supplementary text, to preserve the flow and length of the article.

**5: Major Comment #4**

**Reviewer comment:**

The detailed presentation of streamflow simulation performance (Section 3.1.2) is robust, but its inclusion as a standalone subsection immediately following the core Soil Moisture evaluation (3.1.1) is structurally misleading and requires clarification. Based on the Methods section (Lines 249–257), the streamflow analysis serves primarily as an internal calibration and performance check of the hydrological modeling framework, not a direct validation of the primary variable of interest (soil moisture). Therefore, the detailed streamflow analysis (Section 3.1.2) should be relocated and significantly condensed. This content belongs logically in a dedicated subsection within the Methods (e.g., Model Calibration) to briefly demonstrate the adequacy of the modeling framework, rather than occupying a prominent position in the main Results section. If the authors insist on keeping the streamflow results prominent, they must establish a clear logical link showing how the successful streamflow calibration improves or validates the soil moisture simulations.

**Author response:**

We respectfully note that the manuscript already states that soil moisture and streamflow

are intrinsically linked through the catchment water balance (Methods, Section 2.2.3, Lines 251–269). The streamflow evaluation is therefore presented as an independent consistency check of the simulated water balance, which implicitly includes soil moisture storage and depletion processes. The section is included to add credibility to the modelled soil moisture results. On the suggestion to relocate and condense the section, we can move this section to the Supplementary text and present summary results in Section 3.1.1 of the main article.

**6: Major Comment #5**

**Reviewer comment:**

While Section 3.2 effectively uses the Total Drought Magnitude (TDM) index to characterize the evolution of drought events and identifies a compelling increase in frequency and severity post-2011, the claims of 'three distinct drought regimes' and a 'significant shift' are primarily descriptive, relying heavily on the narrative of the top ten events (Figure 5). This conclusion lacks robust, decade-spanning quantitative evidence. The author should consider providing a concise table or figure reporting key summary statistics for each complete decade (e.g., 1971–1980, 1981–1990, etc.), such as the total cumulative TDM or the mean annual number of drought days. This quantitative evidence is essential to lend statistical rigor to the core finding of the significant shift.

**Author response:**

We would like to note that the manuscript already includes a dedicated statistical decadal analysis in Sections 3.3 and 3.4. This section provides the decadal-scale quantitative evidence supporting the identification of the distinct drought regimes. This includes cumulative drought duration (months per decade), drought severity composition, spatial drought exposure, and bootstrapped confidence intervals for decadal means (Figures 6–8), which show that 2011–2020 is statistically distinct from preceding decades. We acknowledge that this evidence is presented later in the Results and is not explicitly referenced in Section 3.2. To ensure the connection is clearer to readers, we will add a brief forward reference in Section 3.2 indicating that the decadal-scale statistical analysis is presented later in the manuscript. We will also include a summary table reporting key decadal metrics to make the regime shift more immediately apparent.

**7: Major comment #6**

**Reviewer comment:**

The central finding that SMI exhibits stronger persistence and SPEI underestimates deficits (L448, L455) is crucial but currently relies on qualitative descriptions (e.g., 'more persistent,' 'underestimate'). It is better to supplement the discussion with quantified persistence metrics.

For example, for the three major events analyzed (1975–1977, 2016–2017, 2018–2019), the author should consider reporting the following metrics: (a) the maximum duration (in months) of the moderate drought for SMI, SPEI-1, and SPEI-3; and (b) the total number of months under moderate drought for each index during the event period. These data are essential for empirically validating the conclusion that SMI has higher inertia than the precipitation-based indices.

**Author response:**

We thank the reviewer for this comment. We agree that explicit quantitative persistence metrics will strengthen this comparison. We would like to point out that some of the metrics suggested by the reviewer (e.g. maximum drought duration in months) are already reported in Section 3.2 (Lines 363-372). Since this only covers the SMI droughts, we will include these metrics for the SPEI drought.

**8: Major comment #7**

**Reviewer comment:**

The Discussion section could be strengthened by elaborating on the broader implications of this work. How can the findings be applied to larger regions, or integrated into operational drought monitoring and management strategies?

**Author response:**

We do agree with the reviewer on this point. We will add more depth to the Discussion to incorporate the suggestions.

**9: Minor comment**

**Reviewer comment:**

The content in Section 2.2.1 regarding the lack of long-term in situ soil moisture data in Belgium and the subsequent expansion of the model domain is more appropriate for Section 2.2.3. The Input data section should focus primarily on datasets used to drive the model. Standardize the capitalization of section titles, as the current usage is inconsistent. Please standardize the terminology and use 'in situ' consistently throughout the manuscript; some occurrences currently use 'in-situ'. While the term 'Drought Persistence' is used in the text, the definition corresponds more closely to cumulative drought exposure or duration. The Conclusion should focus more on synthesizing key findings and scientific contributions, and discussion-type content should be reduced. Please standardize the terminology and use 'root-zone' consistently throughout the manuscript. It is recommended to present the other drought indices, currently discussed in the Discussion section, within the Methods section.

**Author response:**

Thank you for this important comment. We believe it will improve the readability of the manuscript. We will implement all the suggestions in the revision of our manuscript. We will correct the terminology to avoid confusing the reader.

**10: Special comments**

**Reviewer comment:**

- Line 36: Please correct 'agricultural' to 'agricultural'.

- Line 160: Please remove the extraneous semicolon ';'.

- Line 425 and Line 429: Please correct 'Figure ??(a)' and 'Figure ??(b)'.

- Line 501: Correct 'cosysytems' to 'ecosystems'.

**Author response:**

Thank you for pointing out these errors. We have corrected typographical and formatting issues, and explicitly specified that temperature refers to air temperature. We however note that the misspelling 'cosysytems' does not appear in the current manuscript; the word 'ecosystems' is already spelled correctly.

**References**

Crow, Wade T and Eric F Wood (1999). "Multi-scale dynamics of soil moisture variability observed during SGP'97". In: *Geophysical Research Letters* 26.23, pp. 3485–3488.

Peng, Jian, Alexander Loew, Olivier Merlin, and Niko EC Verhoest (2017). "A review of spatial downscaling of satellite remotely sensed soil moisture". In: *Reviews of Geophysics* 55.2, pp. 341–366.

Wang, Aihui, Dennis P Lettenmaier, and Justin Sheffield (2011). "Soil moisture drought in China, 1950–2006". In: *Journal of Climate* 24.13, pp. 3257–3271.

---

## Author Comment (AC4)

**Author Responses to Reviewer #1 Comments**

Katoria Lekarkar*

We would like to thank the reviewer for the time and effort to review our manuscript. The comments provided by the reviewer will undoubtedly improve our manuscript. In the sections that follow, we provide our responses to individual reviewer comments on the manuscript. The responses to individual comments are written in this **color**.

**1: Overall assessment**

**Reviewer comment:**

The manuscript reconstructs multi-decadal root-zone soil moisture over Belgium with mHM and characterizes drought events using an SMI-based framework, comparing them with precipitation-based indicators (SPEI-1/3). The central finding—that 2011–2020 is the driest decade since at least 1971—is relevant for Belgian drought monitoring and well aligned with broader European trends.

However, the presentation is currently hard to follow due to (i) too many metrics without a clear hierarchy for ranking drought severity. (ii) Several methodological elements (MPR, KDE→SMI percentiles, Fisher-z, NSE definition, event splitting/merging rules) need one-line clarifications so readers can reproduce and interpret results.

**Author response:**

We appreciate this constructive assessment. We agree that the manuscript would benefit from clearer guidance on how drought severity is ranked and from concise clarifications of several methodological components. In the revised manuscript, we will (i) explicitly define a hierarchy of drought-severity metrics, (ii) add short clarifying sentences where methods are introduced, and (iii) improve the structure of the Results section to enhance readability and reproducibility.
* * *
*Corresponding author: katoria.lesaalon.lekarkar@vub.be

**2: Comment #1**

**Reviewer comment:**

The title can be read as if only 2011–2020 is analyzed, yet the study reconstructs 1970–2020 and concludes that 2011–2020 is the driest decade of the five. Please revise the title.

**Author response:**

Thank you for this constructive comment. We will revise the title so that it is clear the study spans the five decades from 1970.

**3: Comment #2**

**Reviewer comment:**

Provide a simple, explicit severity ranking protocol. At present the Results toggle between TDM, SMI$\leq \tau$ area, exposure months, peak area, duration, SPEI-1/3, etc., without a decision rule. Readers cannot tell which event is "most severe." Please state a clear hierarchy.

Add a Table listing the top events with: TDM, peak area (%), duration, exceptional-class exposure.

Annotate TDM values directly in Fig. 5 and state in the caption which tie-breaker decided final ranks when two events are similar.

**Author response:**

We agree that at present the hierarchy of drought classification is not very clear and that can confuse the reader. In the revised manuscript, we will explicitly state that Total Drought Magnitude (TDM) is the primary metric used to rank the droughts as it integrates the spatial extent, duration, and severity into a single measure that enables the comparison of different drought events. We will also establish how the metrics are related.

We will implement these suggestions to improve clarity;

- Add a table listing the top-ranked drought events with TDM, peak affected area, duration, and exceptional-drought exposure;

- Annotate TDM values directly in Figure 5;

- State in the caption how ties or near-equal events are handled (using TDM as the primary ranking criterion).

**4: Comment #3**

**Reviewer comment:**

Lines 363-365, is 2016–2017 or 1975–1977 drought bigger? And based on which indicators? Lines 411-414, based on drought persistence, 2011-2020 is the biggest one.

**Author response:**

This ambiguity arises because 1975–1977 and 2016–2017 are individual drought events. Looking at individual drought events, they are the two biggest droughts. In Lines 411-414, we are referring to the frequency (in months) of droughts in individual decades. So we mean that droughts were most frequent in the 2011-2020 decade. This frequency combines the occurrence of all the droughts in a particular decade. (For 2011-2020 for example, it includes the droughts in 2011, 2016–17, 2018–19, plus all the other smaller droughts). We understand that the language can be confusing to the reader. In the revised manuscript, we will make the language clearer to avoid this confusion and as we propose in comment # 2, we will rank the individual droughts to make the distinction clearer.

**5: Comment #4**

**Reviewer comment:**

Lines 365 and 367, when you talk about area percentage, provide the figure reference.

**Author response:**

Thank you for this suggestion. We will provide the references in the main text so the reader is not lost.

**6: Comment #5**

**Reviewer comment:**

Lines 368–372 discuss 2022–2023, but your decadal analysis ends in 2020. Please remove or move to Discussion/SI with an explicit caveat.

Also line 341 says Fig. 5 covers 1970–2023, but the figure appears to show 1970–2020—please make the figure and caption consistent with the text.

**Author response:**

We agree. We will move the discussion of the 2022–2023 drought to the Discussion, since its inclusion is only to provide additional context. We will also edit Figure 5 accordingly.

**7:  Comment #6**

**Reviewer comment:**

Add a short description about each subsection at the start of Results. Two–three sentences will prevent readers from getting lost.

**Author response:**

Thank you for this suggestion. Indeed it is needed to improve readability and we will add the needed contextual information to each subsection.

**8:  Comment #7**

**Reviewer comment:**

Lines 196-199, explain why resolutions differ.

**Author response:**

As described in Section 2.2, mHM distinguishes between Level-0 (L0) datasets, which define the static morphological datasets (e.g. land use, soils, DEM), and Level-2 (L2) datasets, which represent the meteorological inputs. Since gridded meteorological inputs are often available at coarser resolutions than morphological data, mHM allows the two datasets to be provided in different resolutions. We have explained this in Lines 158–170. The model harmonizes the data internally using the Multiscale Parameter Regionalization (MPR) technique. With MPR, fields of parameters at a given modelling scale are obtained by upscaling their corresponding estimates at the scale of the input data, based on either the arithmetic mean, the geometric or harmonic means, or the majority operator (Kumar et al., 2013). This upscaling leads to quasi scale-invariant parameters that enable mHM to preserve the spatial variability of state variables, conserve mass balance and reduce overparameterization.

We will make this explanation clear in the manuscript.

**9:  Comment #8**

**Reviewer comment:**

Define NSE on first use (Results 3.1.2). Give the range and interpretation ($\approx 1$ perfect; $\approx 0$ equals mean-flow benchmark; $< 0$ worse than mean).

**Author response:**

Thank you for this suggestion. In the revision, we will define NSE and its ranges accordingly.

**10:   Comment #9**

**Reviewer comment:**

Lines 470-475, restate the minimum overlap area rule used to merge adjacent months into one multi-temporal event. Or how do you define duration. This clarifies whether a brief wet interlude (e.g., March–April 2017) splits or does not split an event.

**Author response:**

Indeed. We will make this clear in the revision.

**11:   Comment #10**

**Reviewer comment:**

MPR, what is the full name.

**Author response:**

MPR is an abbreviation for Multiscale Parameter Regionalization. It appears in line 164 of the manuscript.

**12:   Comment #11**

**Reviewer comment:**

Terminology clarity. The manuscript uses three different concepts that contain the word "calibration":

- A 5-year warm-up: is it 1965–1969, if yes, why exclude 1970 as a calibration year for drought analysis?

- Excluding 1970 as a calibration year for drought analysis when forming decades (hence using 1971–1980),

- Streamflow parameter calibration (2000–2023) vs validation (1970–1999).

Please clarify these three meanings to avoid confusion.

**Author response:**

We agree that the current use of the term "calibration" may cause confusion. The warm up period is indeed 1965–1969. The exclusion of 1970 from the drought analysis is not related to model calibration, but to the construction of the SPEI time series. SPEI is based on accumulated water-balance anomalies over preceding months. As a result, January 1970 does not contain valid SPEI-1 values, and January–March 1970 do not contain valid SPEI-3 values. The year 1971 is therefore the first year with complete SPEI values for all months. For this reason, decades are defined starting from 1971 (i.e. 1971–1980). On reflection, referring to 1970 as a calibration year is misleading. We will make the distinction clear in the manuscript.

**References**

Kumar, Rohini, Luis Samaniego, and Sabine Attinger (2013). "Implications of distributed hydrologic model parameterization on water fluxes at multiple scales and locations". In: *Water Resources Research* 49.1, pp. 360–379.